# A Novel Algorithm for Structural Reliability Analysis Based on Finite Step Length and Armijo Line Search

**Peng Huang** [1,2]**, Hong-Zhong Huang** [1,2,*] **and Tudi Huang** [1,2]

[1]    School of Mechanical and Electrical Engineering, University of Electronic Science and Technology of China, Chengdu 611731, China; huangpeng@std.uestc.edu.cn (P.H.); huangtudi@std.uestc.edu.cn (T.H.)
[2]    Center for System Reliability and Safety, University of Electronic Science and Technology of China, Chengdu 611731, China
*    Correspondence: hzhuang@uestc.edu.cn; Tel.: +86-28-6181252

**Abstract:** This paper presents a novel algorithm for structural reliability analysis based on the finite step length and Armijo line search to remove the drawbacks of the Hasofer–Lind and Rakwitz–Fiessler (HL-RF) algorithm that may be subjected to non-convergence in the first-order reliability method (FORM). Initially, the sensitivity factor with finite step length is introduced for preventing the iterative process of the algorithm from entering a periodic loop. Subsequently, an optimization method based on the sufficient descent condition with the Armijo line search technique is proposed. With that, the initial step length and adjusting coefficient are optimized to enhance the applicability of the algorithm emphatically for highly nonlinear functions. A comparison analysis is carried out between the proposed algorithm and existing FORM-based algorithms to validate the robustness and efficiency of the proposed algorithm. The results of this demonstrate that the proposed algorithm is superior to the HL-RF algorithm in terms of robustness and surpass the other existing FORM-based algorithms in connection to efficiency.

**Keywords:** reliability analysis; first-order reliability method; HL-RF algorithm; finite step length; Armijo line search

---

## 1. Introduction

Traditional structural analysis process generally starts with the establishment of reasonable mechanical models, and then obtains the responses based on the deterministic structural parameters and certain external loads [1,2]. However, a number of uncertainties of involved parameters are unneglectable in practical engineering, which may lead the results of reliability analysis to considerable deviations [3–5]. To assess these uncertainties, the probability-based reliability analysis methods spring up and then are maturely applied to a number of sectors and societies [6–9].

The main work in structural reliability analysis is to obtain the failure probability by solving the following multi-dimensional integration:

$$P_f = \Pr\{g(\boldsymbol{X}) < 0\} = \int\limits_{g(\boldsymbol{X})<0} f_X(\boldsymbol{X})dX \tag{1}$$

where $\Pr\{\cdot\}$ is the probability, $\boldsymbol{X} = (X_1, X_2, \cdots, X_n)$ are the basic random variables, $g(\boldsymbol{X})$ is the limit state function (LSF) and $f_X(X)$ is the joint probability density function (PDF) of $\boldsymbol{X}$. Generally, it is arduous to calculate the multiple integral in Equation (1) directly. The commonly used methods are approximate methods and simulation methods [10]. The approximate methods include the first-order reliability method (FORM) [11,12] and the second-order reliability method (SORM) [13,14], which

are defined under the concept of the first and second-order Taylor expansion of the LSF, respectively. The simulation methods, such as direct Monte-Carlo simulation (MCS) [15,16], line sampling [17], and importance sampling [18,19] evaluate the reliability through a considerable amount of stochastic simulations. In engineering practice, the FORM is used the most because of its simplicity and the good balance between accuracy and efficiency.

The Hasofer–Lind and Rakwitz–Fiessler (HL-RF) algorithm, which is proposed by Hasofer and Lind and then is extended by Rakwitz and Fiessler [11,12], is a classical and widely used algorithm in the FORM. However, it may converge in a low speed or even fail to converge any more under the situation that highly nonlinear LSFs are involved. This limitation inspires researchers to devote their efforts to develop algorithms both in academia and the engineering practice sectors. Liu et al. [20] developed a modified HL-RF (MHL-RF) algorithm by introducing a stepsize parameter. The merit function, based on the augmented Lagrange method, is used to find the proposed stepsize. Zhang et al. [21] constructed a simpler merit function; the proposed algorithm named as iHL-RF is superior to the MHL-RF algorithm in efficiency. Santos et al. [22] established a differentiable merit function, and the Wolfe condition is implemented to search the stepsize. Santosh et al. [23] presented a stepsize search method for the MHL-RF algorithm based on the variant of Goldstein's rule. These stepsize determinations have increased a lot the computational effort.

Yang [24] employed chaotic dynamics analysis to construct a stability transformation method (STM) for removing the chaos phenomenon of the HL-RF algorithm. The convergence rate, however, proved to be slow. Keshtegar [25] proposed the chaotic conjugate STM algorithm based on the conjugate gradient method. Meng et al. [26] established a directional STM algorithm using the chaos feedback control approach. Keshtegar et al. [27] presented a relaxed HL-RF (RHL-RF) algorithm by introducing the relaxed coefficient, which is dynamically calculated by the relaxed approach. As a continuation of the research, Meng et al. [28] developed the adaptive STM and enhanced adaptive STM algorithms by transforming the calculation of the most probable failure points into finding the most probable target points. An inappropriate control factor may lead to inefficiency of these algorithms.

To improve the robustness of the HL-RF algorithm, Gong and Yi [29] put forward a finite step length (FSL) algorithm, although with a low computational efficiency as a consequence of inflexible step lengths. Then, efforts were made by Roudak et al. [30] and Keshtegar et al. [31] to improve the convergence rate of partial functions by finding a better choice of the step length of the FSL algorithm. Meanwhile, Keshtegar [32] proposed a conjugate HL-RF (CHL-RF) algorithm based on the conjugate gradient direction, and some relevant works were also given [33,34]. The CHL-RF algorithm can generally improve the convergence rate, but its performance depends strongly on the choice of the conjugate gradient factor and step length. In addition, there are other algorithms similar to the FORM, such as the first-order saddlepoint approximation methods [35], HLRF-BFGS algorithm [36], and hybrid particle swarm optimization algorithm [37].

Robustness and efficiency of the iterative algorithms are the main indexes of the FORM. Benefitting from simplicity and high efficiency, the HL-RF algorithm is the most frequently selected for general reliability analysis. LSFs of structures, however, tend to be increasingly high-dimensional with highly nonlinear functions as a consequence of structure complexity and the scale incremental in an incredible speed, which proved to result in the HL-RF algorithm failing to converge any more. The improved algorithms, e.g., STM, HLRF-BFGS, and RHL-RF algorithms, are alternative options with low efficiency, indicating that the mentioned developed algorithms are still limited in aspects of low robustness and poor efficiency. Motivated by this, this paper puts forward a new algorithm, namely, adaptive finite step length (AFSL), for structural reliability analysis based on the FSL algorithm and the Armijo line search method. In terms of the developed algorithm, a step length together with an Armijo-line-search based optimization method are introduced to improve the robustness and guarantee the efficiency of the algorithm, respectively. Besides, the initial step length and adjusting coefficient are optimized to ensure applicable of the algorithm in nonlinear scenarios.

The structure of the paper is organized as follows. Section 2 presents the proposed AFSL algorithm. Numerical analysis of the robustness and efficiency of the developed algorithms is illustrated in Section 3. Section 4 is the discussions, and the conclusions appear at the end.

## 2. Proposed Adaptive Finite Step Length Algorithm

### 2.1. HL-RF Algorithm

Due to its simplicity and efficiency, the HL-RF algorithm [20] is a widely used algorithm to estimate the failure probability. It includes the following three main steps.

(1) Transform the random variables $X$ into standard normal random variables $U$ by Rosenblatt or Nataf transformation techniques [26], which is expressed as

$$U = \Phi^{-1}[F_X(X)] \tag{2}$$

where $\Phi^{-1}$ is the inverse cumulative distribution function (CDF) of the standard normal distribution, $F_X$ is the CDF of $X$, and $U = (U_1, U_2, \cdots U_n)$.

(2) Find the most probable point (MPP) $U^*$ by using the following iterative formulation:

$$U^{k+1} = \frac{G(U^k) - \nabla^{\mathrm{T}} G(U^k) U^k}{\left\| \nabla G(U^k) \right\|} \alpha^k \tag{3}$$

where $\|\cdot\|$ is the norm of a vector, $k$ is the counter of iteration, $\nabla G(U^k)$ is the gradient of $G(U^k)$, and $\alpha^k$ is the sensitivity factor that can be calculated by

$$\alpha^k = -\frac{\nabla G(U^k)}{\left\| \nabla G(U^k) \right\|} \tag{4}$$

(3) Estimate the failure probability by

$$P_f \approx \Phi(-\beta) \tag{5}$$

where $\Phi$ is the CDF of the standard normal distribution and $\beta = \|U^*\|$ is the reliability index.

However, the HL-RF algorithm may fail to converge for highly nonlinear functions. As shown in Figure 1, assume that $U^k$ is obtained after the iteration of step $k$, $U^{k+1}$ is determined based on the sensitivity factor $\alpha^k$ and the limit state surface. It is obviously that the line $L_1$ is parallel to the negative gradient direction at point $U^k$. If the negative gradient direction at point $U^{k+1}$ is parallel to the line $L_2$, we have $U^{k+2} = U^k$ and the iterative process is caught in a periodic loop. Several improved algorithms, such as the STM [24], HLRF-BFGS [36], and RHL-RF [27] algorithms, have been developed to overcome the non-convergence of HL-RF algorithm, but more computational efforts are needed.

### 2.2. Adaptive Finite Step Length Algorithm

To make the algorithm robust and efficient, this paper introduces a step length parameter into the sensitivity factor. As shown in Figure 2, the sensitivity factor is defined as

$$\alpha^k = \frac{U_a^k}{\left\| U_a^k \right\|} \tag{6}$$

where $U_a^k$ is the auxiliary point along with the negative gradient direction at point $U^k$, which can be calculated by

$$U_a^k = U^k - \lambda \nabla G(U^k) \tag{7}$$

where $\lambda > 0$ is the step length.

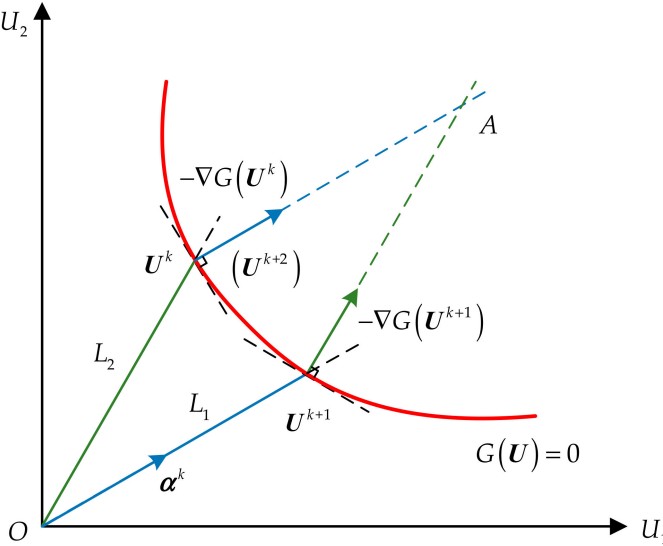

**Figure 1.** Illustration of the periodic oscillation of the Hasofer–Lind and Rakwitz–Fiessler (HL-RF) algorithm.

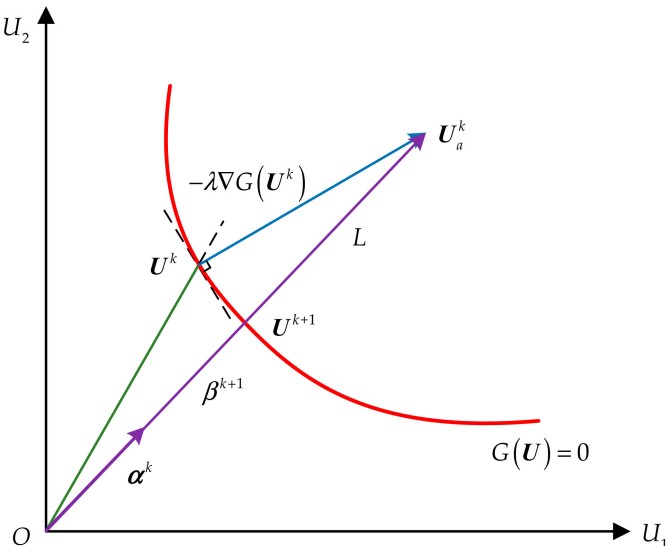

**Figure 2.** Iterative process of the sensitivity factor with step length.

The new point $U^{k+1}$, which is the intersection point of line $L$ and limit state surface, can be obtained by $U^{k+1} = \beta^{k+1}\alpha^k$. In practical engineering, the LSFs of structures are generally too complex to solve directly the reliability index $\beta^{k+1}$ by $G(\beta^{k+1}\alpha^k) = 0$. On this basis, the first-order Taylor expansion of the LSFs is used, and $G(\beta^{k+1}\alpha^k) = 0$ is approximated as

$$G(U^k) + \nabla^{\mathrm{T}}G(U^k)(\beta^{k+1}\alpha^k - U^k) \approx 0 \tag{8}$$

The reliability index is accordingly computed as

$$\beta^{k+1} \approx \frac{\nabla^{\mathrm{T}}G(U^k)U^k - G(U^k)}{\nabla^{\mathrm{T}}G(U^k)\alpha^k} \tag{9}$$

Consequently, the iterative formulation can be summarized as

$$u^{k+1} = \frac{\nabla^{\mathrm{T}} G(u^k) u^k - G(u^k)}{\nabla^{\mathrm{T}} G(u^k) \alpha^k} \alpha^k \tag{10}$$

where,

$$\alpha^k = \frac{u^k - \lambda \nabla G(u^k)}{\left\| u^k - \lambda \nabla G(u^k) \right\|} \tag{11}$$

Noted that if $\lambda \to \infty$, Equation (11) becomes

$$\lim_{\lambda \to \infty} \alpha^k = \lim_{\lambda^k \to \infty} \frac{u^k/\lambda - \nabla G(u^k)}{\left\| u^k/\lambda - \nabla G(u^k) \right\|} = -\frac{\nabla G(u^k)}{\left\| \nabla G(u^k) \right\|} \tag{12}$$

Equation (12) indicates that the HL-RF algorithm is a special case of the above iterative formulation with infinite step length. Therefore, it is named as FSL algorithm [29].

Compared with the HL-RF algorithm, the same simple FSL algorithm is better in robustness. It does not need the extra computational effort to calculate the step length. However, the step length is hard to determine properly for various nonlinear LSFs. For example, if the LSF is highly nonlinear, a large step length may make the algorithm fail to converge; otherwise, the convergence rate may be slow. To address these issues, an adaptive optimization method is proposed for FSL algorithm based on the sufficient descent condition with the Armijo line search in this paper.

According to the sufficient descent condition [33], the adjustment of step length is expressed as

$$\lambda^{k+1} = \begin{cases} \lambda^k & \left\| u^{k+1} - u^k \right\| < \left\| u^k - u^{k-1} \right\| \\ c\lambda^k & \left\| u^{k+1} - u^k \right\| \geq \left\| u^k - u^{k-1} \right\| \end{cases} \tag{13}$$

where $c$ is the adjusting coefficient, and a recommended value for $c$ in this paper is between 0.5~0.6.

In Equation (13), if $\left\| u^{k+1} - u^k \right\| \geq \left\| u^k - u^{k-1} \right\|$, the step length is set as $\lambda^{k+1} = c\lambda^k$. In the extreme case that $\left\| u^{k+1} - u^k \right\| \gg \left\| u^k - u^{k-1} \right\|$, $u^{k+1}$ is the point with a large deviation; this deviation will make the calculation inefficient, especially when the point nears the MPP. Therefore, such iteration points need to be optimized. The optimization model is defined as

$$u_*^{k+1} = u^k + \theta^k d^k \tag{14}$$

where $u_*^{k+1}$ is the optimized iteration point of $u^{k+1}$, $\theta^k$ is the optimized coefficient, and $d^k$ is the search direction that given by

$$d^k = u^{k+1} - u^k \tag{15}$$

Theoretically, the optimized coefficient should minimize the value of the LSF along the search direction. However, it requires the exact line search to result in considerable computational effort [38–40]. For most optimization algorithms, the convergence rate does not depend on the exact search process. Therefore, the inexact line search can not only guarantee the acceptable reduction of the objective function but also make the final iteration sequence convergence, indicating which is powerful. The Armijo line search method [31] is defined as

$$G\left(u^k + \beta^b d^k\right) \leq G\left(u^k\right) + \sigma \beta^b \nabla^{\mathrm{T}} G\left(u^k\right) d^k \tag{16}$$

where $\sigma, \beta \in (0, 1)$ are pre-selected parameters, $b$ is the smallest nonnegative integer satisfying the inequality, and $\theta = \beta^b$.

To ensure the global convergence, the merit function proposed by Zhang and Kiureghian [21] is used, which is given as

$$m(\pmb{U}) = \frac{1}{2}\|\pmb{U}\| + \rho|G(\pmb{U})| \tag{17}$$

where $\rho$ is a constant and should satisfy $\rho > \|\pmb{U}\|/\nabla G(\pmb{U})$.

According to Equations (16) and (17), the optimized coefficient $\theta^k$ can be calculated by

$$\theta^k = \max_b\left\{\beta^b\middle|m\left(\pmb{U}^k + \beta^b\pmb{d}^k\right) - m\left(\pmb{U}^k\right) < 0\right\} \tag{18}$$

where $\beta = 0.5$ and $\rho = \|\pmb{U}\|/\nabla G(\pmb{U}) + 10$ are used in the proposed method.

In addition, the initial step length is generally selected as 50 in the FSL algorithm [29], which is unsuitable to a number of LSFs. In this paper, the initial step length is empirically defined as

$$\lambda^0 = \min\left(\frac{50}{\left\|\nabla G\left(\pmb{U}^0\right)\right\|}, 50\right) \tag{19}$$

where $\pmb{U}^0$ is the initial point, and the mean point in standard normal space is the same as the initial point.

The convergence properties of the AFSL algorithm are analyzed as follows. Existing a step length $\lambda^k$ makes $\left\|\pmb{U}^{k+1} - \pmb{U}^k\right\| < \left\|\pmb{U}^k - \pmb{U}^{k-1}\right\|$. Let $\left\|\pmb{U}^{k+1} - \pmb{U}^k\right\| = t^k\left\|\pmb{U}^k - \pmb{U}^{k-1}\right\|$ with $0 < t^k < 1$, we have

$$\left\|\pmb{U}^{k+1} - \pmb{U}^k\right\| = t^k\left\|\pmb{U}^k - \pmb{U}^{k-1}\right\| = t^kt^{k-1}\left\|\pmb{U}^{k-1} - \pmb{U}^{k-2}\right\| = \cdots = \prod_{i=1}^{k}t^i\left\|\pmb{U}^1 - \pmb{U}^0\right\| \tag{20}$$

where $\lim_{k\to\infty}\prod_{i=1}^{k}t^i \approx 0$. Therefore, $\lim_{k\to\infty}\left\|\pmb{U}^{k+1} - \pmb{U}^k\right\| \approx 0$, showing the convergence of the proposed algorithm.

In addition, according to Equation (7), we have

$$\frac{U_i^{k+1}}{U_j^{k+1}} = \frac{\alpha_i^k}{\alpha_j^k} = \frac{U_i^k - \lambda^k\dfrac{\partial G\left(\pmb{U}^k\right)}{\partial U_i}}{U_j^k - \lambda^k\dfrac{\partial G\left(\pmb{U}^k\right)}{\partial U_j}} \tag{21}$$

where $i, j = 1, 2, \cdots, n$ and $i \neq j$.

Knowing that $\lim_{k\to\infty}\left\|\pmb{U}^{k+1} - \pmb{U}^k\right\| \approx 0$, Equation (21) can be modified as

$$\frac{\lim\limits_{k\to\infty}U_i^{k+1}}{\lim\limits_{k\to\infty}U_j^{k+1}} = \frac{\lim\limits_{k\to\infty}\alpha_i^k}{\lim\limits_{k\to\infty}\alpha_j^k} = \frac{-\dfrac{\partial G\left(\pmb{U}^k\right)}{\partial U_i}}{-\dfrac{\partial G\left(\pmb{U}^k\right)}{\partial U_j}} \tag{22}$$

Therefore,

$$\lim_{k\to\infty}\pmb{\alpha}^k = -\frac{\nabla G\left(\pmb{U}^k\right)}{\left\|\nabla G\left(\pmb{U}^k\right)\right\|} \tag{23}$$

The vector $\pmb{\alpha}^\infty$ is orthogonal to the limit state surface, $\pmb{U}^\infty$ is the MPP. According to the above, the proposed algorithm can find the MPP easily.

*2.3. Summary of the Proposed Algorithm*

Three features of the proposed AFSL algorithm for reliability analysis are concluded as: (i) inheritance, that is, the proposed algorithm process features of the FSL algorithm are as simple as those of the HL-RF algorithm but superior in robustness; (ii) efficiency, the proposed algorithm is more efficient than the FSL algorithm due to the iteration point adaptive optimization method based on the Armijo line search being employed; (iii) availability, the AFSL algorithm is suitable for various nonlinear LSFs by establishing the relationship between the initial step length and the initial iteration point. Consequently, the proposed algorithm can efficiently deal with various reliability analysis problems, especially for highly nonlinear functions. The framework of the proposed AFSL algorithm is shown in Figure 3, and the main steps are summarized as follows:

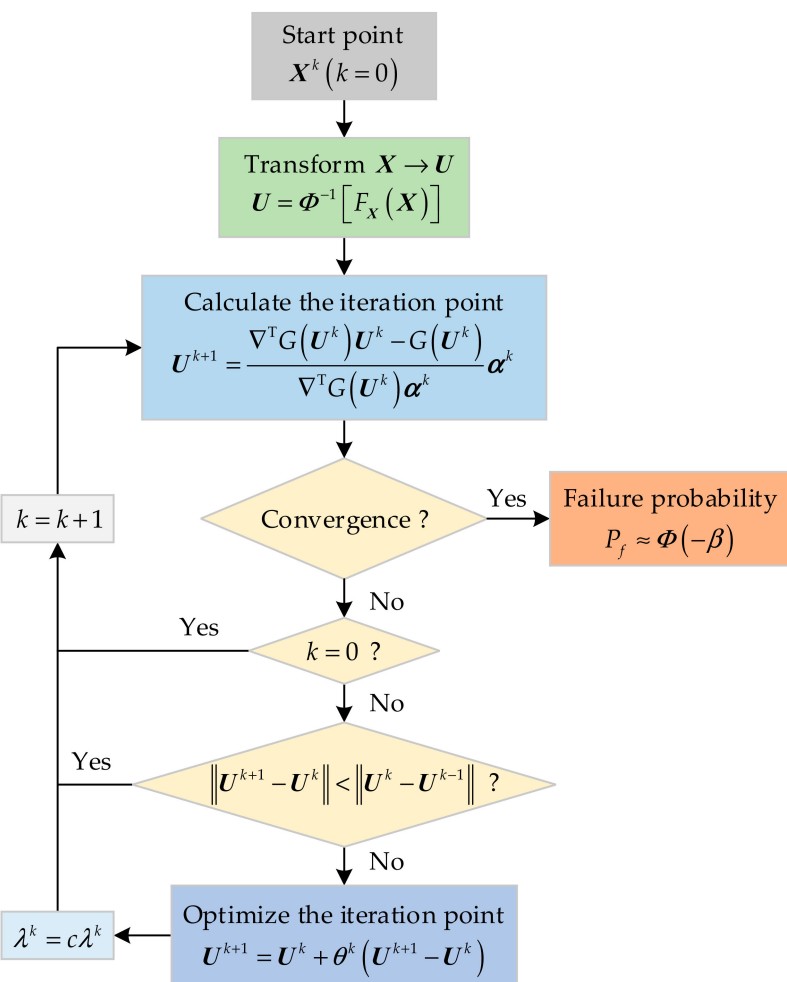

**Figure 3.** Framework of the proposed AFSL algorithm.

(1) Let $k = 0$, and set the start point $X^0$.

(2) Transform the random variables $X$ into standard normal random variables $U$ by Equation (2).

(3) Calculate the sensitivity factor $\alpha^k$ by Equation (11).

(4) Determine the new point $U^{k+1}$ based on Equation (10).

(5) Check convergence. If $\left\| U^{k+1} - U^k \right\| / \left\| U^k \right\| \leq \varepsilon$ ($\varepsilon$ is a small positive number), then stop, $\beta = \left\| U^{k+1} \right\|$ and go to step (8); otherwise, go to step (6).

(6) If $k = 0$, set $k = k + 1$ and go to step (3); otherwise, go to step (7).

(7) If $\left\| U^{k+1} - U^k \right\| < \left\| U^k - U^{k-1} \right\|$, set $k = k + 1$ and go to step (3); otherwise, optimize the iteration point $U^{k+1}$ based on Equation (14), set $\lambda^k = c\lambda^k$ and $k = k + 1$, go to step (3).

(8) The failure probability is estimated by Equation (5).

## 3. Illustrative Examples

Five nonlinear examples are given, in this section, to demonstrate the robustness and efficiency of the proposed AFSL algorithm by comparisons with the HL-RF [20], STM [24] (with $\lambda = 0.1$ and $\boldsymbol{C} = \boldsymbol{I}$), HLRF-BFGS [36], RHL-RF [27], and FSL [29] (with $\lambda = 50$ and $c = 1.5$) algorithms. The parameter in the proposed AFSL algorithm is set as $c = 0.55$. The same stopping criterions ($\varepsilon = 10^{-6}$) are used for all algorithms.

### 3.1. Mathematical Examples

**Example 1.** *A highly nonlinear function is given as [26]*

$$G_1 = X_1^4 + 2X_2^4 - 20 \tag{24}$$

*where $X_1$ and $X_2$ are normal random variables with $\mu_1 = \mu_2 = 10$ and $\sigma_1 = \sigma_2 = 5$.*

The number of iterations, $G$-evaluations, $\nabla G$-evaluations, and reliability index for all algorithms are listed in Table 1. Figure 4 shows the iterative processes of reliability index. According to Table 1, the STM, RHL-RF, FSL, and proposed AFSL algorithms converge to stable solutions, but the HL-RF and HLRF-BFGS algorithms fail to converge. The HL-RF algorithm produces the periodic solutions of {0.9267, 0.9863}. The results calculated by MCS with $10^6$ samples are $P_f = 1.87 \times 10^{-3}$ and $\beta = 2.8998$. The AFSL algorithm converges to the stable result as $\beta = 2.3842$, which is more accurate than the other algorithms. In addition, according to the number of iterations, $G$-evaluations and $\nabla G$-evaluations, they have shown that the proposed algorithm is more efficient than the other algorithms.

**Table 1.** Results of different algorithms for Example 1.

| Algorithms | Iterations | $G$-Evaluations | $\nabla G$-Evaluations | $\beta$ | $P_f$ |
|:---:|:---:|:---:|:---:|:---:|:---:|
| HL-RF | – | – | – | – | – |
| STM | 154 | 154 | 154 | 2.3654 | $9.01 \times 10^{-3}$ |
| HLRF-BFGS | – | – | – | – | – |
| RHL-RF | 181 | 361 | 181 | 2.3654 | $9.01 \times 10^{-3}$ |
| FSL | 108 | 108 | 108 | 2.3655 | $9.00 \times 10^{-3}$ |
| Proposed AFSL | 26 | 47 | 26 | 2.3842 | $8.56 \times 10^{-3}$ |

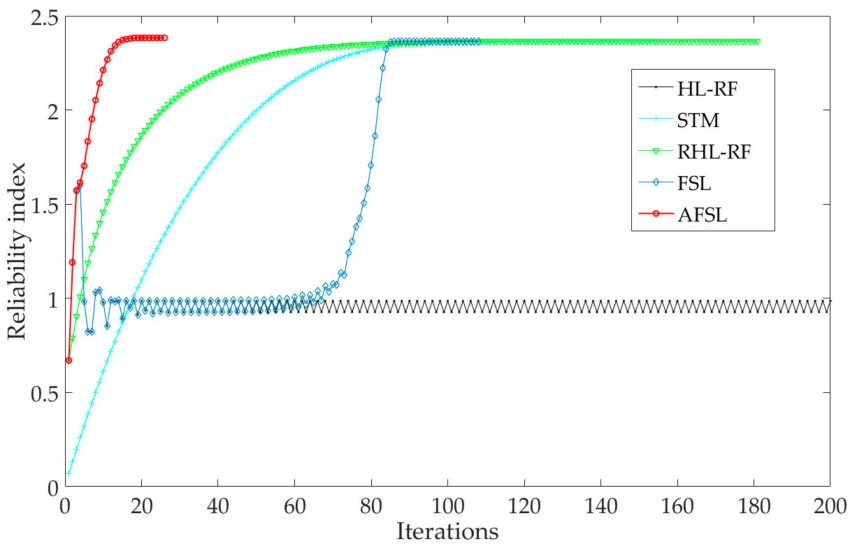

**Figure 4.** Iterative processes of reliability index for Example 1.

**Example 2.** *A convex performance function is given by [29]*

$$G_2 = \frac{1}{P} \ln\{\exp[P(1 + X_1 - X_2)] + \exp[P(5 - 5X_1 - X_2)]\} \tag{25}$$

*where* $P = 1$, *both* $X_1$ *and* $X_2$ *are standard normal variables. Its limit state surface is shown in Figure 5.*

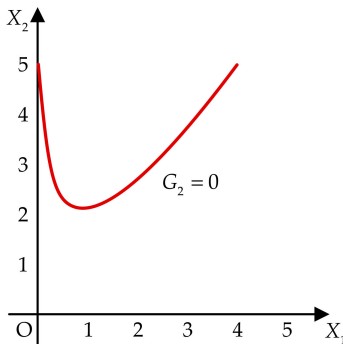

**Figure 5.** The limit state surface for Example 2.

The iterative processes of reliability index for all compared algorithms are given in Table 2. The iterative processes of reliability index are shown in Figure 6. The failure probability computed by MCS with $10^6$ samples is $3.43 \times 10^{-3}$ ($\beta = 2.7035$). From Table 2 and Figure 6, the HL-RF algorithm fails to converge, for its iterative process falls into a periodic loop. The STM, HLRF-BFGS, RHL-RF, FSL, and the proposed AFSL algorithms converge to stable results, i.e., $\beta = 2.2995$, $P_f = 1.07 \times 10^{-2}$ and $X^* = (0.8641, 2.1310)$. The number of iterations is 119, 16, 106 43, and 17, respectively. The proposed AFSL and HLRF-BFGS algorithms are more efficient than the STM, RHL-RF, and FSL algorithms.

**Table 2.** Results of different algorithms for Example 2.

| Algorithms | Iterations | G-Evaluations | ∇G-Evaluations | $\beta$ | $P_f$ |
|:---:|:---:|:---:|:---:|:---:|:---:|
| HL-RF | – | – | – | – | – |
| STM | 119 | 119 | 119 | 2.2995 | $1.07 \times 10^{-2}$ |
| HLRF-BFGS | 16 | 16 | 16 | 2.2995 | $1.07 \times 10^{-2}$ |
| RHL-RF | 106 | 211 | 106 | 2.2995 | $1.07 \times 10^{-2}$ |
| FSL | 43 | 43 | 43 | 2.2995 | $1.07 \times 10^{-2}$ |
| Proposed AFSL | 17 | 26 | 17 | 2.2995 | $1.07 \times 10^{-2}$ |

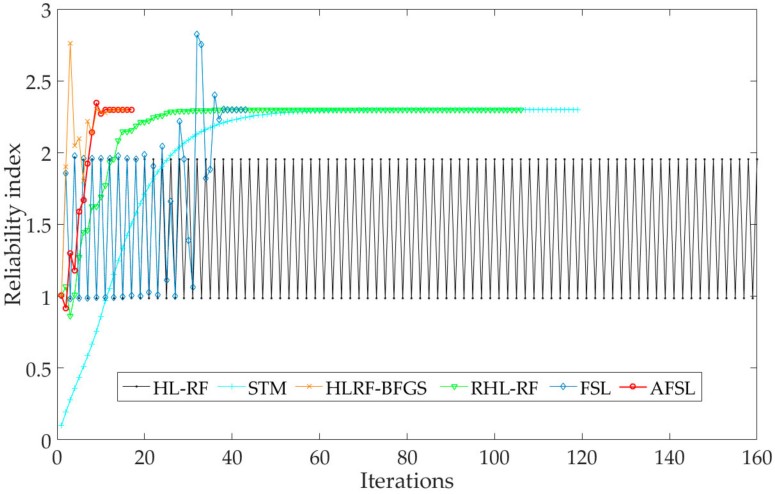

**Figure 6.** Iterative processes of reliability index for Example 2.

### 3.2. Structural/Mechanical Examples

**Example 3.** *A vehicle side-impact with the performance function is expressed as [26]*

$$G_3 = 0.489X_3X_7 + 0.843X_5X_6 - 0.0432X_9X_{10} + 0.0556X_9X_{11} + 0.000786X_{11}^2 - 0.75 \qquad (26)$$

*where $X_i(i = 1, \cdots, 11)$ are normal variables, for $X_1 \sim X_7$, $\mu = 1$, and $\sigma = 0.05$, for $X_8 \sim X_9$, $\mu = 0,3$ and $\sigma = 0.006$, and for $X_{10} \sim X_{11}$, $\mu = 0$, and $\sigma = 10$.*

Results of the HL-RF, STM, HLRF-BFGS, RHL-RF, FSL, and the proposed AFSL algorithms are listed in Table 3. The iterative processes of reliability index are shown in Figure 7. The failure probability based on the MCS with $10^7$ samples is $1.24 \times 10^{-4}$ ($\beta = 3.6643$). According to Table 3 and Figure 7, the reliability indexes of all algorithms converge to 3.4975, except the HL-RF and HLRF-BFGS algorithms. The HL-RF algorithm produces the periodic solutions of {1.2957, 3.1757, 2.0351, 1.9180}. The number of iterations, *G*-evaluations and $\nabla G$-evaluations prove that the proposed AFSL algorithm is more efficient than other algorithms.

**Table 3.** Results of different algorithms for Example 3.

| Algorithms | Iterations | *G*-Evaluations | $\nabla G$-Evaluations | $\beta$ | $P_f$ |
|---|---|---|---|---|---|
| HL-RF | – | – | – | – | – |
| STM | 114 | 114 | 114 | 3.4975 | $2.35 \times 10^{-4}$ |
| HLRF-BFGS | – | – | – | – | – |
| RHL-RF | 120 | 239 | 120 | 3.4975 | $2.35 \times 10^{-4}$ |
| FSL | 64 | 64 | 64 | 3.4975 | $2.35 \times 10^{-4}$ |
| Proposed AFSL | 36 | 39 | 36 | 3.4975 | $2.35 \times 10^{-4}$ |

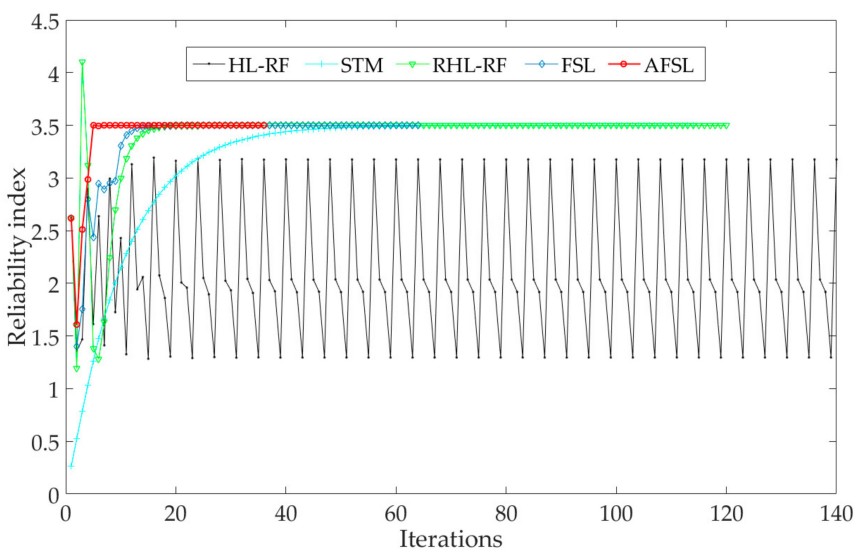

**Figure 7.** Iterative processes of reliability index for Example 3.

**Example 4.** *A ten-bar truss structure is shown in Figure 8. Based on the tip displacement at Node 3, its LSF is given by [27]*

$$G_4 = d_0 - D \qquad (27)$$

*where $d_0 = 0.11(m)$ is the allowable value, and the maximum displacement D is computed by*

$$D = \frac{PL}{A_1 A_2 E} \left\{ \frac{4\sqrt{2}A_1^3(24A_1^2 + A_3^2) + A_3^3(7A_1^2 + 26A_2^2)}{4A_2^2(8A_1^2 + A_3^2) + 4\sqrt{2}A_1 A_2 A_3(3A_1 + 4A_2) + A_1 A_3^2(A_1 + 6A_2)} + \frac{4A_1 A_2 A_3(20A_1^2 + 76A_1 A_2 + 10A_3^2 + 25\sqrt{2}A_1 A_3 + 29\sqrt{2}A_2 A_3)}{4A_2^2(8A_1^2 + A_3^2) + 4\sqrt{2}A_1 A_2 A_3(3A_1 + 4A_2) + A_1 A_3^2(A_1 + 6A_2)} \right\} \quad (28)$$

The relevant parameters and their distribution are described in Table 4.

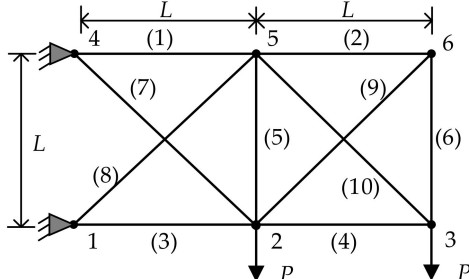

**Figure 8.** Ten-bar truss structure.

**Table 4.** Statistics of random variables in Example 4.

| Variable | Distribution | Mean | Coefficient of Variation | Dimension |
|----------|--------------|------|--------------------------|-----------|
| $A_1$ | Normal | $8.39 \times 10^{-3}$ | 0.05 | $m^2$ |
| $A_2$ | Normal | $1.29 \times 10^{-3}$ | 0.05 | $m^2$ |
| $A_3$ | Normal | $5.81 \times 10^{-3}$ | 0.05 | $m^2$ |
| $P$ | Gumbel | $4.45 \times 10^5$ | 0.1 | N |
| $L$ | Normal | 9.14 | 0.05 | m |
| $E$ | Log-normal | $6.89 \times 10^4$ | 0.05 | MPa |

The results of the HL-RF, STM, HLRF-BFGS, RHL-RF, FSL, and the proposed AFSL algorithms are given in Table 5. It can be seen that all algorithms converge to the point $X^* = (8.30 \times 10^{-3}, 1.29 \times 10^{-3}, 5.75 \times 10^{-3}, 4.85 \times 10^5, 9.33, 6.75 \times 10^4)$ with $P_f = 0.1258$ ($\beta = 1.1464$). The result is close to $P_f = 0.1345$ ($\beta = 1.1055$), which is computed by MCS with $10^5$ samples. The number of iterations of the HL-RF and proposed AFSL algorithms is 7, which is more efficient than the other algorithms in this example.

**Table 5.** Convergent results of different algorithms for Example 4.

| Algorithms | HL-RF | STM | HLRF-BFGS | RHL-RF | FSL | AFSL |
|------------|-------|-----|-----------|--------|-----|------|
| $A_1(m^2)$ | $8.30 \times 10^{-3}$ | $8.30 \times 10^{-3}$ | $8.30 \times 10^{-3}$ | $8.30 \times 10^{-3}$ | $8.30 \times 10^{-3}$ | $8.30 \times 10^{-3}$ |
| $A_2(m^2)$ | $1.29 \times 10^{-3}$ | $1.29 \times 10^{-3}$ | $1.29 \times 10^{-3}$ | $1.29 \times 10^{-3}$ | $1.29 \times 10^{-3}$ | $1.29 \times 10^{-3}$ |
| $A_3(m^2)$ | $5.75 \times 10^{-3}$ | $5.75 \times 10^{-3}$ | $5.75 \times 10^{-3}$ | $5.75 \times 10^{-3}$ | $5.75 \times 10^{-3}$ | $5.75 \times 10^{-3}$ |
| $P(N)$ | $4.85 \times 10^5$ | $4.85 \times 10^5$ | $4.85 \times 10^5$ | $4.85 \times 10^5$ | $4.85 \times 10^5$ | $4.85 \times 10^5$ |
| $L(m)$ | 9.33 | 9.33 | 9.33 | 9.33 | 9.33 | 9.33 |
| $E(MPa)$ | $6.75 \times 10^4$ | $6.75 \times 10^4$ | $6.75 \times 10^4$ | $6.75 \times 10^4$ | $6.75 \times 10^4$ | $6.75 \times 10^4$ |
| $\beta$ | 1.1464 | 1.1464 | 1.1464 | 1.1464 | 1.1464 | 1.1464 |
| $P_f$ | 0.1258 | 0.1258 | 0.1258 | 0.1258 | 0.1258 | 0.1258 |
| Iterations | 7 | 115 | 127 | 20 | 9 | 7 |

**Example 5.** *A two-degree of freedom dynamic system is shown in Figure 9. According to the force capacity of the secondary spring, the LSF is expressed as [34]*

$$G_5 = F_s - K_s \times P\left(E\left[x_s^2\right]\right)^{1/2} \tag{29}$$

*where $F_s$ is the force, $P = 3$ is the peak factor and mean-square relative displacement of the secondary spring $E\left[x_s^2\right]$ is computed by*

$$E\left[x_s^2\right] = \frac{\pi S_0}{4\xi_s\omega_s}\left[\frac{\xi_a\xi_s}{\xi_p\xi_s\left(4\xi_a^2 + \theta^2\right) + \gamma\xi_a^2} \times \frac{\left(\xi_p\omega_p^3 + \xi_s\omega_s^3\right)\omega_p}{4\xi_a\omega_a^4}\right] \tag{30}$$

*where $S_0$ is intensity of the white noise, $\gamma = M_s/M_p$, $\omega_p = \sqrt{K_p/M_p}$, $\omega_s = \sqrt{K_s/M_s}$, $\omega_a = \left(\omega_p + \omega_s\right)/2$, $\xi_a = \left(\xi_p + \xi_s\right)/2$ and $\theta = \left(\omega_p - \omega_s\right)/\omega_a$. The information of eight random variables with lognormal distribution is given in Table 6.*

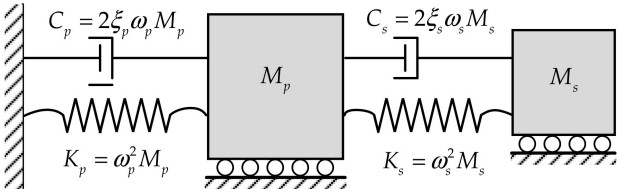

**Figure 9.** Two-degrees of freedom dynamic system.

**Table 6.** Statistics of random variables in Example 5.

| Random Variable | $M_p$ | $M_s$ | $K_p$ | $K_s$ | $\xi_p$ | $\xi_s$ | $F_s$ | $S_0$ |
|---|---|---|---|---|---|---|---|---|
| Mean | 1 | 0.01 | 1 | 0.01 | 0.05 | 0.02 | 15 | 100 |
| Standard Deviation | 0.1 | 0.001 | 0.2 | 0.001 | 0.02 | 0.01 | 1.5 | 10 |

The number of iterations, $G$-evaluations, $\nabla G$-evaluations, and reliability index for all algorithms are listed in Table 7. The iterative processes of reliability index are shown in Figure 10. Using the MCS with $10^6$ samples, the failure probability is obtained as $5.04 \times 10^{-3}$ ($\beta = 2.5733$). The HL-RF algorithm has failed to converge. It yields the periodic solutions of {3.2644, 3.6503}. The other algorithms converge to the stable results as $\beta = 2.1002$, $P_f = 1.79 \times 10^{-2}$ and $X^* =$(1.02, 9.98 $\times 10^{-3}$, 1.06, 1.04 $\times 10^{-2}$, 2.67 $\times 10^{-2}$, 1.16 $\times 10^{-2}$, 13.61, 104.19). However, the proposed AFSL algorithm is more efficient than the other algorithms.

**Table 7.** Results of different algorithms for Example 5.

| Algorithms | Iterations | $G$-Evaluations | $\nabla G$-Evaluations | $\beta$ | $P_f$ |
|---|---|---|---|---|---|
| HL-RF | – | – | – | – | – |
| STM | 187 | 187 | 187 | 2.1002 | $1.79 \times 10^{-2}$ |
| HLRF-BFGS | 294 | 294 | 294 | 2.1002 | $1.79 \times 10^{-2}$ |
| RHL-RF | 144 | 287 | 144 | 2.1002 | $1.79 \times 10^{-2}$ |
| FSL | 256 | 256 | 256 | 2.1002 | $1.79 \times 10^{-2}$ |
| Proposed AFSL | 82 | 94 | 82 | 2.1002 | $1.79 \times 10^{-2}$ |

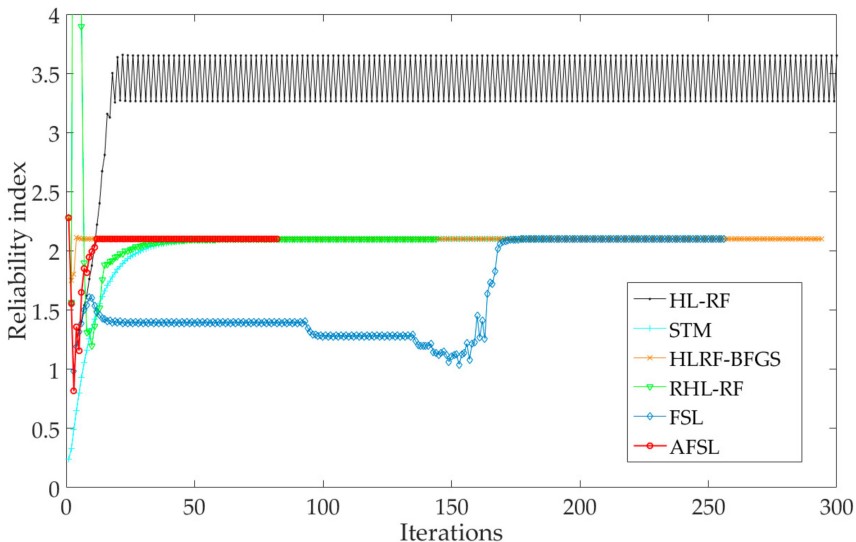

**Figure 10.** Iterative processes of reliability index for Example 5.

## 4. Discussions

In the previous section, five nonlinear examples were used to prove the performance of the proposed AFSL algorithm in both effectiveness and convergence aspects, by comparison the calculated reliability index and number of iterations with the HL-RF, STM, HLRF-BFGS, RHL-RF, and FSL algorithms are used for comparisons in each example. The results globally are listed in Table 8. It can be concluded that the HL-RF algorithm has the worst robustness. The HL-RF algorithm converges only in Example 4. In Examples 1–3 and 5, its iterative processes are trapped in a periodic loop. The HLRF-BFGS algorithm fails to converge in Examples 1 and 3. Furthermore, its convergence rate is the lowest in Examples 4 and 5. Other improved HL-RF algorithms, i.e., STM, RHL-RF, and FSL algorithms, converge to stable results in all examples. However, they are inefficient. According to the convergent results of Examples 1–5, the proposed AFSL algorithm is more robust than the HL-RF and HLRF-BFGS algorithms, and more efficient than the STM, RHL-RF, and FSL algorithms. Consequently, the proposed AFSL algorithm has the best convergence performance.

**Table 8.** Convergent results (i.e., reliability index\iterations) for Examples 1–5.

| Algorithms | Example 1 | Example 2 | Example 3 | Example 4 | Example 5 |
|---|---|---|---|---|---|
| HL-RF | – | – | – | 1.1464\7 | – |
| STM | 2.3654\154 | 2.2995\119 | 3.4975\114 | 1.1464\115 | 2.1002\187 |
| HLRF-BFGS | – | 2.2995\16 | – | 1.1464\127 | 2.1002\294 |
| RHL-RF | 2.3654\181 | 2.2995\106 | 3.4975\120 | 1.1464\20 | 2.1002\144 |
| FSL | 2.3655\108 | 2.2995\43 | 3.4975\64 | 1.1464\9 | 2.1002\256 |
| Proposed AFSL | 2.3842\26 | 2.2995\17 | 3.4975\36 | 1.1464\7 | 2.1002\82 |

Noted that the proposed AFSL algorithm is developed based on the FSL algorithm. To illustrate its advantages, a further comparison is made between the FSL algorithm and the proposed AFSL algorithm with the same parameters ($\lambda = 35$ and $c = 1.4$), as tabulated in Table 9. In Example 4, the numbers of $G$ and $\nabla G$ evaluations are the same for that the iteration points are non-adjustable. However, in Examples 1–3 and 5, the numbers of $G$ and $\nabla G$ evaluations have reduced significantly. Figure 11 shows the details of the iterative processes for Example 3. Therefore, the computational effort is effectively decreased by the proposed optimization method. This adaptive optimization method in the iterative process can also be applied to other similar algorithms.

**Table 9.** Comparisons (i.e., *G*-evaluations\∇*G* -evaluations) of the FSL and AFSL algorithms.

| Algorithms | Example 1 | Example 2 | Example 3 | Example 4 | Example 5 |
|---|---|---|---|---|---|
| FSL | 121\121 | 55\55 | 184\184 | 9\9 | 290\290 |
| Proposed AFSL | 114\48 | 50\36 | 32\29 | 9\9 | 113\82 |

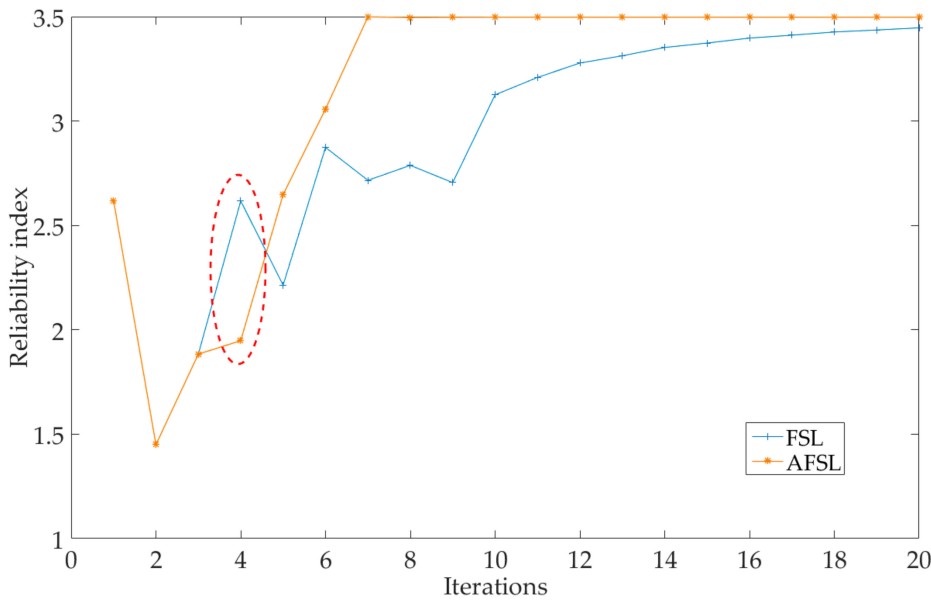

**Figure 11.** The comparisons of iterative process for Example 3.

The selection of adjusting coefficient in the proposed AFSL algorithm is studied as well. Figure 12 shows the number of iterations of the proposed AFSL algorithm with different adjusting coefficients. It can be seen that the adjusting coefficient $c$ does not have significant effects on the iterations of the proposed AFSL algorithm. For the recommended value of $c \in [0.5, \ 0.6]$, the proposed AFSL algorithm can converge effectively.

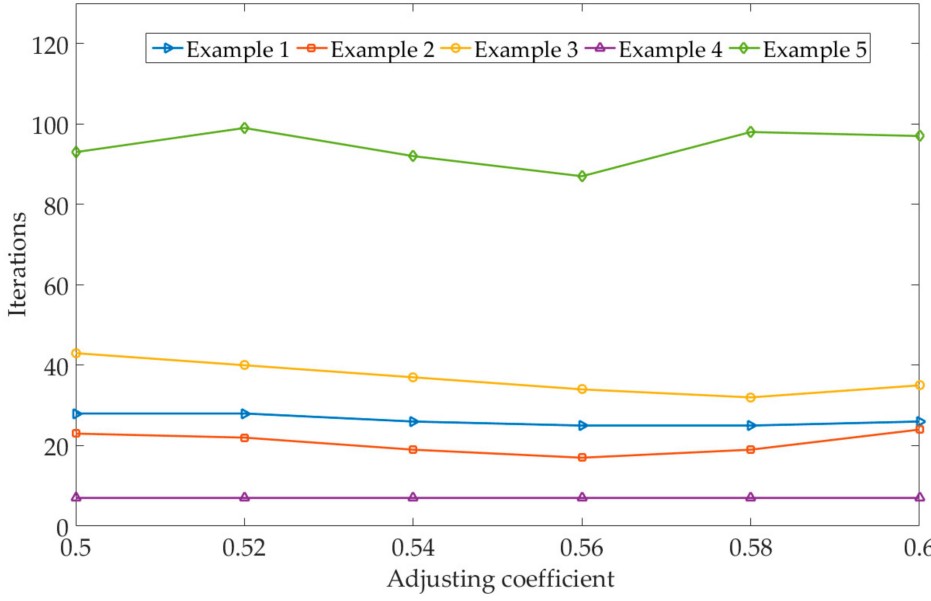

**Figure 12.** Iterations of the proposed AFSL algorithm with different adjusting coefficients.

## 5. Conclusions

In this paper, a robust and efficient algorithm for structural reliability analysis, namely, the AFSL algorithm is developed. The algorithm can solve the problems like the non-convergence issue for the HL-RF algorithm and the inefficiency of several existing FORM-based algorithms.

Firstly, a step length parameter is introduced to the HL-RF algorithm which avoids the iterative process from falling into periodic loop and then makes the proposed algorithm more robust. Secondly, an adaptive iteration point optimization method is proposed based on the sufficient descent condition with the Armijo line search, aiming at optimizing the iteration points with large errors in the iterative process by which the proposed algorithm can be more efficient than many existing FORM-based algorithms. Finally, the initial step length and the adjusting coefficient are optimized, making the proposed algorithm more suitable for various nonlinear functions. The robustness and efficiency of the proposed AFSL algorithm are concluded from the results comparison with several existing FORM-based algorithms, e.g., the HL-RF, STM, HLRF-BFGS, RHL-RF, and FSL algorithms. The robustness and efficiency of the proposed algorithm is verified by five selected functions from the published literature, and those functions are receiving attention in both academia and engineering practice.

However, it should be mentioned that the accuracy of the proposed algorithm may be hard to guarantee for highly nonlinear functions. The error is mainly caused by the first-order Taylor approximation of the LSF, which is an inherent problem of the FORM. Therefore, more accurate methods, such as the advanced line sampling and importance sampling, should be combined to improve the accuracy of the proposed algorithm in the future.

**Author Contributions:** Conceptualization, P.H. and H.H.; data curation, P.H. and T.H.; formal analysis, P.H. and H.H.; methodology, P.H.; validation, P.H. and T.H.; writing—original draft preparation, P.H.; writing—review and editing, H.H. and T.H.

**Funding:** This research was funded by the National Key R&D Program of China (Grant No. 2017YFB1301300).

**Conflicts of Interest:** The authors declare no conflicts of interest.

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
