# Peer review of "A Novel Algorithm for Structural Reliability Analysis Based on Finite Step Length and Armijo Line Search"

_applsci, doi:10.3390/app9122546_

Round 1

Reviewer 1 Report

Section 2 can be found in any textbook and has to be removed.

The authors should outline that they pursue an improvement where the gradient of the objective function is available. Today genetic algorithms allows to afford any kind of objective function, see Probabilistic Engineering Mechanics, 2014, 36, 72-80, 2014

Author Response

Point 1: Section 2 can be found in any textbook and has to be removed. 

Response 1: Thanks for the comment. As suggested, section 2 has been removed in our revised version.

Point 2: The authors should outline that they pursue an improvement where the gradient of the objective function is available. Today genetic algorithms allow to afford any kind of objective function, see Probabilistic Engineering Mechanics, 2014, 36, 72-80, 2014.

Response 2: Thanks for the comment. Genetic algorithm is effective in resolving the optimization problems. However, its computational efficiency is low for a system with time-consuming implicit functions and large number of variables. Therefore, gradient-based reliability method is used because its convergence speed is fast and it provides a good balance between accuracy and efficiency.

Reviewer 2 Report

In the manuscript the authors proposed to use Armijo line search to remove some of the  drawbacks of the Hasofer-Lind and Rakwitz-Fiessler, known as HLRF algorithm for structural reliability analysis. In particular the authors proposed the used of the algorithm to estimate the design point for highly non-linear systems. 

The authors should also clearly state the advantages and differences with  of the  approach presented in this manuscript against existing works proposed to improve the HLRF algorithms (e.g. https://doi.org/10.1016/j.probengmech.2015.09.012 and https://doi.org/10.1016/j.apm.2014.10.024).  

However there is a much more important research question to be answered. In theory this approach can prevent the HLRF algorithm to enter in an infinite loop and then not being able to converge. At first, this seems a big limitation. However, it is well known that first order reliability method works very well under two conditions:

* Linear or quasi linear limit state function

* Low dimensional space. 

The fact that HLRF algorithm does not work well when the above conditions are not satisfied it is not a limitation. On the contrary, this can be an advantage because it prevents researches to perform FORM on such systems. It well known that FORM is not applicable and should not be applied for those systems (see e.g. https://doi.org/10.1016/j.strusafe.2009.08.004 ) and therefore the applicability of the proposed approach to real problem is questionable. 

The design point and reliability index in non-linear high dimensional systems is not informative (and in fact the authors did not report the values of probability of failure but only the reliability index). 

Possible applications of the proposed approach is when it is combined with advanced monte carlo methods.  For instance Advanced Line Sampling, Structural safety, 2015, 52 Part B, 170-182 ( https://doi.org/10.1016/j.strusafe.2014.10.002 ) required the use of important direction (that can be based on the design point). Hence, this approach might be of importance for such approach. 

Also the Importance sampling for parallel systems, Structural safety 2011, 33(1), 1-7

(https://doi.org/10.1016/j.strusafe.2010.04.002)  requires the calculation of design points or intersection among failure domains and this approach can be adopted to calculate directly the required point without the need of linearisation of individual limit state functions.  
The above mentioned algorithms are available as part of the opencossan project. 

Therefore the authors should reconsider and reposition their research. 

Other remarks

* All the units in the example proposed must be in the international metric system.   

Author Response

Response to Reviewer 2 Comments

Point 1: In the manuscript the authors proposed to use Armijo line search to remove some of the drawbacks of the Hasofer-Lind and Rakwitz-Fiessler, known as HLRF algorithm for structural reliability analysis. In particular the authors proposed the used of the algorithm to estimate the design point for highly non-linear systems.

The authors should also clearly state the advantages and differences with of the approach presented in this manuscript against existing works proposed to improve the HLRF algorithms (e.g. https://doi.org/10.1016/j.probengmech.2015.09.012 and https://doi.org/10.1016/j.apm.2014.10.024).

Response 1: Thanks for the comment. For the first paper (https://doi.org/10.1016/j.probengmech.2015.09.012), the author proposed a method for computing the SORM factor, it may be unfair to compare this method with FORM-based algorithms. Therefore, the comparison with this method is not carried out. The HLRF-BFGS algorithm, as given in the second paper (https://doi.org/10.1016/j.apm.2014.10.024), is employed for comparative analysis in our revised version. The corresponding changes are in blue (see Tables 1-3, 5 and 7). According to the results, the HLRF-BFGS algorithm fails to converge in Examples 1 and 3. Furthermore, the convergence rate of the HLRF-BFGS algorithm is the lowest in Examples 4 and 5. Therefore, the proposed algorithm generally has a better convergence performance than others.

Table 1. Results of different algorithms for Example 1.

Algorithms

Iterations

-evaluations

-evaluations

HL-RF

-

-

-

-

-

STM

154

154

154

2.3654

9.01×10-3

HLRF-BFGS

-

-

-

-

-

RHL-RF

181

361

181

2.3654

9.01×10-3

FSL

108

108

108

2.3655

9.00×10-3

Proposed AFSL

26

47

26

2.3842

8.56×10-3

Table 2. Results of different algorithms for Example 2

Algorithms

Iterations

-evaluations

-evaluations

HL-RF

-

-

-

-

-

STM

119

119

119

2.2995

1.07×10-2

HLRF-BFGS

16

16

16

2.2995

1.07×10-2

RHL-RF

106

211

106

2.2995

1.07×10-2

FSL

43

43

43

2.2995

1.07×10-2

Proposed AFSL

17

26

17

2.2995

1.07×10-2

Table 3. Results of different algorithms for Example 3

Algorithms

Iterations

-evaluations

-evaluations

HL-RF

-

-

-

-

-

STM

114

114

114

3.4975

2.35×10-4

HLRF-BFGS

-

-

-

-

-

RHL-RF

120

239

120

3.4975

2.35×10-4

FSL

64

64

64

3.4975

2.35×10-4

Proposed AFSL

36

39

36

3.4975

2.35×10-4

Table 5. Convergent results of different algorithms for Example 4

Algorithms

HL-RF

STM

HLRF-BFGS

RHL-RF

FSL

AFSL

(m2)

8.30×10-3

8.30×10-3

8.30×10-3

8.30×10-3

8.30×10-3

8.30×10-3

(m2)

1.29×10-3

1.29×10-3

1.29×10-3

1.29×10-3

1.29×10-3

1.29×10-3

(m2)

5.75×10-3

5.75×10-3

5.75×10-3

5.75×10-3

5.75×10-3

5.75×10-3

(N)

4.85×105

4.85×105

4.85×105

4.85×105

4.85×105

4.85×105

(m)

9.33

9.33

9.33

9.33

9.33

9.33

(MPa)

6.75×104

6.75×104

6.75×104

6.75×104

6.75×104

6.75×104

1.1464

1.1464

1.1464

1.1464

1.1464

1.1464

0.1258

0.1258

0.1258

0.1258

0.1258

0.1258

Iterations

7

115

127

20

9

7

Table 7. Results of different algorithms for Example 5

Algorithms

Iterations

-evaluations

-evaluations

HL-RF

-

-

-

-

-

STM

187

187

187

2.1002

1.79×10-2

HLRF-BFGS

294

294

294

2.1002

1.79×10-2

RHL-RF

144

287

144

2.1002

1.79×10-2

FSL

256

256

256

2.1002

1.79×10-2

Proposed AFSL

82

94

82

2.1002

1.79×10-2

Point 2: However there is a much more important research question to be answered. In theory this approach can prevent the HLRF algorithm to enter in an infinite loop and then not being able to converge. At first, this seems a big limitation. However, it is well known that first order reliability method works very well under two conditions:

* Linear or quasi linear limit state function

* Low dimensional space.

The fact that HLRF algorithm does not work well when the above conditions are not satisfied it is not a limitation. On the contrary, this can be an advantage because it prevents researches to perform FORM on such systems. It well known that FORM is not applicable and should not be applied for those systems (see e.g. https://doi.org/10.1016/j.strusafe.2009.08.004) and therefore the applicability of the proposed approach to real problem is questionable.

Response 2: As the reviewer pointed out, the limitation of FORM-based algorithms that mentioned above are really exist. However, in engineering cases, it is difficult to determine the nonlinear degree of limit state function, especially for a system with implicit limit state function. Therefore, the robustness and accuracy of the FORM cannot be predicted. If the HL-RF algorithm is used, it may has endless loop or produce unstable result. However, the proposed algorithm can address these problems properly. In addition, many methods with high accuracy are based on the FORM-based algorithms to calculate the reliability index and search the most probable point, such as the SORM and the important sampling method. Therefore, the proposed algorithm with a better robustness and efficiency has its engineering application value.

Point 3: The design point and reliability index in non-linear high dimensional systems is not informative (and in fact the authors did not report the values of probability of failure but only the reliability index).

Response 3: We listed all required probability of failure in this version, see Tables 1-3, 5 and 7 (as shown in Response 1).

Point 4: Possible applications of the proposed approach is when it is combined with advanced monte carlo methods. For instance Advanced Line Sampling, Structural safety, 2015, 52 Part B, 170-182 (https://doi.org/10.1016/j.strusafe.2014.10.002) required the use of important direction (that can be based on the design point). Hence, this approach might be of importance for such approach. Also the Importance sampling for parallel systems, Structural safety 2011, 33(1), 1-7 (https://doi.org/10.1016/j.strusafe.2010.04.002) requires the calculation of design points or intersection among failure domains and this approach can be adopted to calculate directly the required point without the need of linearisation of individual limit state functions.

The above mentioned algorithms are available as part of the opencossan project. Therefore the authors should reconsider and reposition their research.

Response 4: Thanks for comment. Simulation methods are widely used in structural reliability analysis for its high accuracy. However, its disadvantage is the low computational efficiency. If the advanced Monte Carlo methods (e.g. advanced line sampling and importance sampling) can be applied together with the proposed algorithm, the accuracy of the proposed algorithm may be improved and with high efficiency. The corresponding methods have been discussed in revised version. See references below and in the revised manuscript:

5.  Goller, B., Pradlwarter, H.J., Schueller, G. I. Reliability assessment in structural dynamics. J. Sound Vibr. 2013, 332, 2488-2499.

9.  Valdebenito, M.A.; Pradlwarter, H.J.; Schueller, G.I. The role of the design point for calculating failure probabilities in view of dimensionality and structural nonlinearities. Struct. Saf. 2010, 32, 101-111.

13.  Patelli, E.; Feng, G.; Coolen, F.P.; Coolen-Maturi, T. Simulation methods for system reliability using the survival signature. Reliab. Eng. Syst. Saf. 2017, 167, 327-337.

14.  Angelis, M.D.; Patelli, E.; Beer, M. Advanced line sampling for efficient robust reliability analysis. Struct. Saf. 2015, 52, 170-182.

15.  Patelli, E.; Pradlwarter, H.J.; Schueller, G.I. On multinormal integrals by importance sampling for parallel system reliability. Struct. Saf. 2011, 33, 1-7.

16.  Beaurepaire, P.; Jensen, H.A.; Schuller, G.I.; Valdebenito, M.A. Reliability-based optimization using bridge importance sampling. Probab. Eng. Eng. Mech. 2013, 34, 48-57.

Point 5: All the units in the example proposed must be in the international metric system.

Response 5: Thanks for the comment. We have revised all the units accordingly.

Round 2

Reviewer 1 Report

Authors' answer is not the expected, but the manuscript could be accepted as it is.

Author Response

Response: Thanks for the comment and recommendation.

Reviewer 2 Report

The authors have provided a significant improved version of the originally submitted paper.
The limitation of the proposed approach and on the FORM method in general are clearly stated. 

It would be good to know the true value of the probability of failure of the provided examples. For instance, in the example 1 the author stated that MCS with 10^6 is used to estimate the reliability index of 2.8998. However, I believe the MCS is used to calculate the probability of failure not the reliability index. These two quantities coincide only for linear state.

Therefore the values of the probability of failure computed via Monte Carlo should also be reported

Author Response

Response: The results from the Monte Carlo simulation for all  examples have been reported in our revised version and the corresponding results are in blue in this revised version.